# MOVIS: Enhancing Multi-Object Novel View Synthesis for Indoor Scenes

## Abstract

Repurposing pre-trained diffusion models has been proven to be effective for novel view synthesis (NVS). However, these methods are mostly limited to a single object; directly applying such methods to compositional multi-object scenarios yields inferior results, especially incorrect object placement and inconsistent shape and appearance under novel views. How to enhance and systematically evaluate the cross-view consistency of such models remains under-explored. To address this issue, we propose MOVIS to enhance the structural awareness of the view-conditioned diffusion model for multi-object NVS in terms of model inputs, auxiliary tasks, and training strategy. First, we inject structure-aware features, including depth and object mask, into the denoising U-Net to enhance the model's comprehension of object instances and their spatial relationships. Second, we introduce an auxiliary task requiring the model to simultaneously predict novel view object masks, further improving the model's capability in differentiating and placing objects. Finally, we conduct an in-depth analysis of the diffusion sampling process and carefully devise a structure-guided timestep sampling scheduler during training, which balances the learning of global object placement and fine-grained detail recovery. To systematically evaluate the plausibility of synthesized images, we propose to assess cross-view consistency and novel view object placement alongside existing image-level NVS metrics. Extensive experiments on challenging synthetic and realistic datasets demonstrate that our method exhibits strong generalization capabilities and produces consistent novel view synthesis, highlighting its potential to guide future 3D-aware multi-object NVS tasks.

## 1 Introduction

Novel view synthesis (NVS) from a single image is imperative for various applications, including AR/VR, interior designs, robotics, *etc.* This is highly challenging as it requires understanding complex spatial structures from a single 2D perspective observation while being able to extrapolate consistent and plausible content for unobserved areas. The substantial demands for comprehensive knowledge of the 3D world render it a difficult task, even for humans with rich priors of the 3D environments.

Recently, significant progress has been made in the realm of single-object image-to-3D generation (Tang et al., 2023b; Liu et al., 2023b; Shi et al., 2023a; Liu et al., 2023a; Shi et al., 2023b; Long et al., 2024) empowered by the advances in 2D diffusion models (Rombach et al., 2022; Ho et al., 2020). Among them, one prominent line of research (Liu et al., 2023b; Lin et al., 2023a; Qian et al., 2023; Tang et al., 2023a; Weng et al., 2023; Lin et al., 2023b; Liu et al., 2023d; Huang et al., 2023; Chen et al., 2023) has achieved compelling results by building on insights from Zero-1-to-3 (Liu et al., 2023c): repurposing a pre-trained diffusion model as a novel view synthesizer by fine-tuning on large 3D object datasets can provide promising 3D-aware prior for image-to-3D tasks.

However, these methods are mostly restricted to the single-object level. It remains unclear if this paradigm can be effectively extended to the multi-object level to facilitate more complex tasks like reconstructing an indoor scene. In Fig. 1, we visualize cross-view matching results of directly applying the aforementioned novel view synthesizers (Liu et al., 2023c) in multi-object scenarios, which showcases weak consistency with input views. Specifically, we believe that the lack of structural awareness is the primary reason for the disappearance, distortion, incorrect position, and orientation of objects under novel views. While several works (Sargent et al., 2023; Tung et al., 2024)

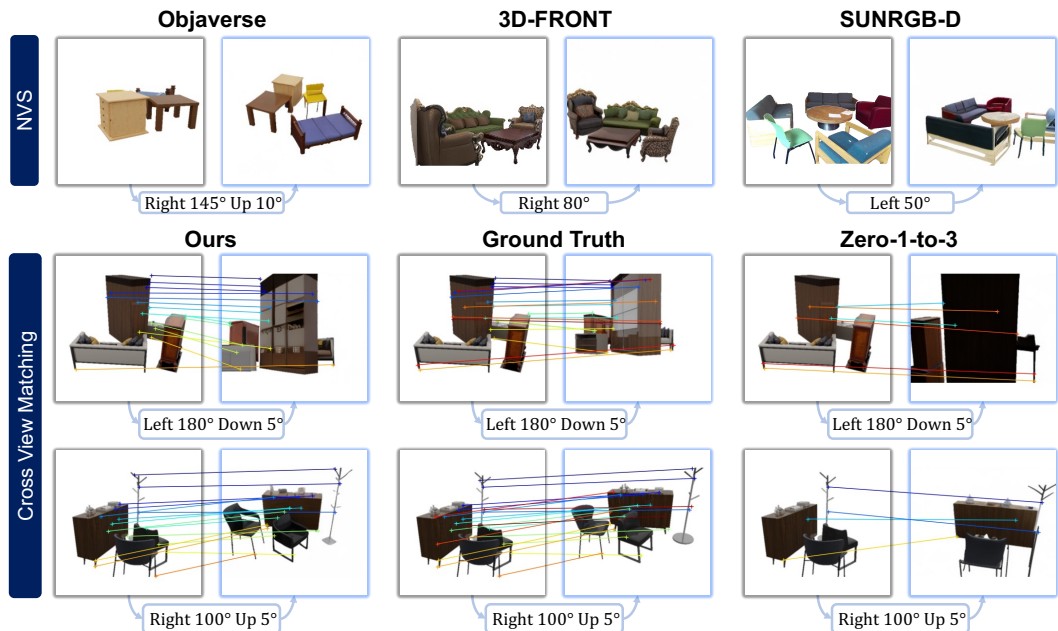

Figure 1: **Novel view synthesis and cross-view image matching**. The first row shows that MOVIS generalizes to different datasets on NVS. We also show visualizations of cross-view consistency compared with Zero-1-to-3 (Liu et al., 2023c) and ground truth by applying image-matching. MOVIS can match a significantly greater number of points, closely aligned with the ground truth.

have explored training on mixed real-world scene datasets, the complexity introduced by multiple objects, such as spatial placement, per-instance geometry and appearance, and occlusion relationship, makes incorporating such awareness non-trivial.

Inspired by the discussion above, our paper seeks to address the question: *How to enhance the structural awareness of current diffusion-based novel view synthesizers?* We begin by identifying the key challenges in extending single-object methods for multi-object NVS tasks. A multi-object image possesses more complicated structural information than a single-object one. The model must first grasp the hierarchical structure within, which includes both high-level global object placement, *e.g.*, position and orientation, and low-level ones like per-object geometry and appearance. High-level structural information significantly reduces the ambiguity in object composition while low-level details are essential for accurately capturing the characteristics of each object instance. Subsequently, the model needs to retain this hierarchical information captured from the input view while synthesizing novel-view images to ensure cross-view structural consistency. These capabilities are less critical in single-object level NVS tasks due to the reduced ambiguity in one-to-one mapping but are crucial for effective multi-object NVS models.

Building on these insights, our technical designs are threefold. We first propose injecting structure-aware features, *i.e.*, depth and object mask, from the input view as additional inputs to provide information on both high-level global placement and fine-grained local details. Secondly, we utilize the prediction of novel view object masks as an auxiliary task during training for the model to differentiate object instances, laying a solid foundation for fine-grained geometry and appearance recovery. Finally, through an in-depth analysis of the model's inference process, we highlight the importance of revising the noise timestep sampling schedule, which influences the learning focus in the training process. To be specific, larger timesteps emphasize global placement learning, while smaller timesteps focus on local fine-grained object geometry and appearance recovery. To endow the view-conditioned diffusion model with both capabilities, we propose a structure-guided timestep sampling scheduler that prioritizes larger timesteps in the initial stage, gradually decreasing over time to balance these two conflicting inductive biases. This design is fundamental to our proposed model's effectiveness in addressing the complexity of multi-object level NVS tasks.

To systematically assess the plausibility of synthesized novel view images, we additionally evaluate novel-view object mask and cross-view structural consistency apart from the existing NVS metrics. Specifically, we employ image-matching techniques (Wang et al., 2024; Leroy et al., 2024) to compare the input-view image with both the ground-truth and synthesized novel-view images. Cross-view structural consistency evaluates how closely the matching results align, providing a measure of the accuracy in recovering object placement, shape, and appearance. On the other hand, the object mask, as measured by Intersection over Union (IoU), assesses the precision of object placement. Extensive experiments demonstrate that our method excels at multi-object level NVS in indoor scenes, achieving consistent object placement, shape, and appearance. Notably, it exhibits strong generalization capabilities for generating novel views on unseen datasets, including both synthetic ones 3D-FRONT (Fu et al., 2021a), Room-Texture (Luo et al., 2024) and Objaverse (Deitke et al., 2023b), as well as the real-world SUNRGB-D (Song et al., 2015).

In summary, our paper focuses on enhancing structural awareness of view-conditioned diffusion models, improving the quality and consistency of synthesized images. Our main contributions are:

1. We introduce structure-aware features as model inputs and incorporate novel view mask prediction as an auxiliary task during training. This enhances the model's understanding of hierarchical structures in multi-object scenarios, leading to improved NVS performance.
2. We present a novel noise timestep sampling scheduler designed to balance the learning of global object placement and fine-grained detail recovery, which is critical for addressing the increased complexity in multi-object scenarios.
3. We introduce additional metrics to systematically evaluate the novel view structural consistency. Through extensive experiments, our model demonstrates superiority in consistent object placement, geometry, and appearance recovery, showcasing strong generalization capability to unseen datasets.

## 2 RELATED WORK

### 2.1 SINGLE OBJECT NVS WITH GENERATIVE MODELS

Synthesizing novel view images for single objects given a single-view image is an extremely ill-posed problem that requires strong priors. With great advances achieved in diffusion models (Ho et al., 2020; Rombach et al., 2022), research efforts (Xu et al., 2023; Tang et al., 2023b; Melas-Kyriazi et al., 2023) seek to distill priors (Jain et al., 2022; Poole et al., 2022) learned from Text-to-Image (T2I) diffusion models via image captioning like Li et al. (2023). However, this presents a huge gap between the image and semantics due to the ambiguity of the text, hindering the 3D consistency of these methods. On the other hand, view-conditioned diffusion models like Zero-1-to-3 (Liu et al., 2023c) explore an Image-to-Image (I2I) generation paradigm that "teaches" the diffusion model to control viewpoints to synthesize plausible images under novel views, providing a more consistent 3D-aware prior. Subsequent work focuses on accelerating the generation speed (Liu et al., 2023b; Tang et al., 2023a), enhancing the view consistency (Chen et al., 2023; Lin et al., 2023b; Weng et al., 2023; Liu et al., 2023d; Huang et al., 2023), or accelerating the training process (Jiang et al., 2023). However, all these methods deal with single and complete object novel view synthesis tasks since they usually fine-tune their model on Objaverse (Deitke et al., 2023b;a), an extensive single-object level dataset, contrary to real images which normally consist of multiple or incomplete objects. The lack of specific model designs for compositional scenes also leads to significant inconsistencies when directly applying them to the multi-object scenarios, as can be seen from Fig. 1.

### 2.2 MULTI-OBJECT 3D RECONSTRUCTION WITH SINGLE OBJECT PRIORS

Following the advance in 3D-aware single object generative prior (Liu et al., 2023c; Shi et al., 2023a), a line of research work (Chen et al., 2024b; Dogaru et al., 2024; Chen et al., 2024a) focuses on extending their application to compositional multi-object scenarios. The core idea is to decompose object compositions into individual objects, thereby fully leveraging the powerful generative priors of single-object models. They first break down a multi-object composition into several components via segmentation models like SAM (Kirillov et al., 2023), and then complete every single object with amodal (Ozguroglu et al., 2024; Xu et al., 2024; Zhan et al., 2024) or inpainting (Rombach et al., 2022; Lugmayr et al., 2022) techniques. The object instances are lifted to 3D via image-to-3D models (Tang et al., 2023a; Wu et al., 2024; Liu et al., 2023c; Wang & Shi, 2023) and finally composited into

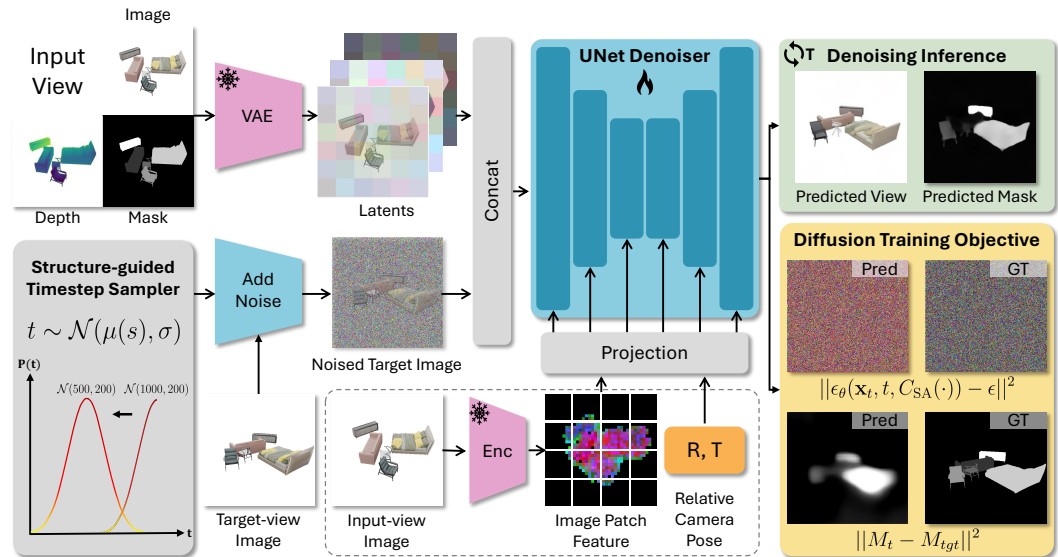

Figure 2: **Overview of MOVIS.** Our model performs NVS from the input image and relative camera change. We introduce structure-aware features as additional inputs and employ mask prediction as an auxiliary task (Sec. 3.2). The model is trained with a structure-guided timestep sampling scheduler (Sec. 3.3) to balance the learning of global object placement and local detail recovery.

a whole utilizing spatial-aware optimization, 3D bounding box detection (Brazil et al., 2023; Nie et al., 2020) or carefully estimating the metric depth (Ke et al., 2024; Yang et al., 2024b). However, this divide-and-conquer paradigm is limited by the user-specified spatial relations from language prompts (Chen et al., 2024b) and relies heavily on the cascaded modules of detection (Kirillov et al., 2023; Brazil et al., 2023; Ke et al., 2024), completion (Rombach et al., 2022; Lugmayr et al., 2022) and 3D-aware object-level novel view synthesis (NVS) (Liu et al., 2023b; Wang & Shi, 2023) to provide priors for reconstruction. Unlike any of the above, our method aims to build an end-to-end image-conditioned novel view synthesis model that can directly cope with the increased complexity in multi-object compositions, especially in indoor scenes with multiple furniture items.

## 2.3 SCENE-LEVEL NVS WITH SPARSE VIEW INPUT

Early efforts (Jain et al., 2021; Yu et al., 2021; Wang et al., 2021) attempted to directly perform scene-level NVS tasks by extracting image features from input-view images and inferring the underlying 3D representation (Mildenhall et al., 2020). With the development of Gaussian Splatting (Kerbl et al., 2023), recent works (Charatan et al., 2023; Chen et al., 2024c) attempt to switch the underlying representation to Gaussian Splatting for efficiency. However, they mainly deal with synthesizing views near input ones with limited generative capabilities to the unseen region. Inspired by the great success of diffusion models (Rombach et al., 2022) and the object-level 3D-aware novel-view synthesizer (Liu et al., 2023c; Wang & Shi, 2023), several recent works have also attempted to perform scene-level NVS tasks by directly conditioning the generative models on a single-view scene image or a monocular dynamic scene video (Van Hoorick et al., 2024). ZeroNVS (Sargent et al., 2023) proposes to train a view-conditioned diffusion model on a mixture of real-world datasets, MegaScenes (Tung et al., 2024) further scales up the training dataset with Internet-level data pairs for stronger generalization capabilities. However, all these works mainly deal with small view-change and simple scenarios in terms of object number, with few adaptations to tackle the multi-object complexity. In this work, we systematically examine the cross-view consistency of NVS by proposing new metrics, and explore the critical designs required to enhance the structural consistency of the view-conditioned diffusion models in the multi-object scenarios.

## 3 METHOD

In this section, we address the challenge of enhancing the structural awareness of diffusion-based novel view synthesizers for better cross-view consistency in multi-object scenarios. We begin with a brief introduction to diffusion models and view-conditioned diffusion models (Sec. 3.1). Next, we detail the key architectural designs of MOVIS, including how we incorporate structural-aware features as input to improve the model's understanding of hierarchical structure information (Sec. 3.2) and how we introduce novel view mask prediction as an auxiliary task, instructing the model to differentiate the object instances with correct object placement (Sec. 3.2). Finally, we provide an in-depth analysis of the inference process and adopt a structure-guided timestep sampling scheduler (Sec. 3.3) to balance the learning of global object placement and local fine-grained object geometry and appearance recovery. We provide an overview of our view-conditioned diffusion model in Fig. 2.

### 3.1 PRELIMINARIES

**Diffusion Models**    Diffusion models learn to generate images by gradually adding noise to an image (forward process) and recovering the original image from a noisy image (backward process) (Ho et al., 2020). Specifically, in the forward process, Gaussian noise is progressively introduced to the image via $q(\mathbf{x}_t|\mathbf{x}_{t-1})$. Due to the additivity of Gaussian distributions, this iterative process can be written as $q(\mathbf{x}_t|\mathbf{x}_0) = \mathcal{N}(\alpha_t\mathbf{x}_0, \sigma_t^2\mathbf{I})$, where $\alpha_t$ and $\sigma_t$ are designed to converge to $\mathcal{N}(\mathbf{0}, \mathbf{I})$ at the end of the forward process (Kingma et al., 2021; Song et al., 2020b). In the backward process, the model learns to progressively denoise from a noisy image $p_\theta(\mathbf{x}_{t-1}|\mathbf{x}_t)$. This learning is formulated as learning the noise estimator $\epsilon_\theta(\mathbf{x}_t, t)$ following Ho et al. (2020):

$$\mathbb{E}[||\epsilon_\theta(\alpha_t\mathbf{x}_0 + \sigma_t\epsilon, t) - \epsilon||_2^2], \tag{1}$$

where $\epsilon$ is drawn from $\mathcal{N}(\mathbf{0}, \mathbf{I})$ and the timestep $t$ is uniformly sampled from $\mathcal{U}(\mathbf{1}, \mathbf{1000})$. In the inference stage, one can either apply a stochastic (Ho et al., 2020) or a deterministic (Song et al., 2020a) sampler to generate high-quality images via iterative refinement.

**View-conditioned Diffusion Models**    Diffusion models have been recently repurposed as a novel view synthesizer. By training on posed image pairs $\{(\mathbf{x}_0, \hat{\mathbf{x}}_0)\}$ where $\hat{\mathbf{x}}_0 \in \mathbb{R}^{H \times W \times 3}$ denotes the input view image and $\mathbf{x}_0 \in \mathbb{R}^{H \times W \times 3}$ denotes the target view, view-conditioned diffusion models (Watson et al., 2022; Liu et al., 2023c) use the input image $\hat{\mathbf{x}}_0$ and camera pose transformation as conditions to predict the target view image $\mathbf{x}_0$ from a different viewpoint. Specifically, the learning objective of view-conditioned diffusion models is:

$$\mathbb{E}[||\epsilon_\theta(\alpha_t\mathbf{x}_0 + \sigma_t\epsilon, t, C(\hat{\mathbf{x}}_0, R, T)) - \epsilon||_2^2], \tag{2}$$

where $R, T$ represent the relative camera pose transformation between the target view $\mathbf{x}_0$ and the input view $\hat{\mathbf{x}}_0$. $C(\hat{\mathbf{x}}_0, R, T)$ is the view-conditioned feature, combining the relative camera pose transformation with encoded image features to form a new 'pose-aware' feature map, taking the place of the origin CLIP (Radford et al., 2021) feature embedding. Moreover, input view image $\mathbf{x}_0$ will be concatenated with the noisy image as the input of the denoising U-Net. As discussed in Sec. 2.1, single-image-based NVS is extremely challenging, current methods inherit natural image priors from large-scale pre-training (Rombach et al., 2022) and fine-tune diffusion models on large-scale 3D object datasets like Objaverse (Deitke et al., 2023b) to learn the transformation between objects in the input and novel views given the relative camera pose. Despite their ability to generalize to in-the-wild objects, these view-conditioned diffusion models struggle with multi-object scenarios like multi-furniture indoor scenes due to the scarcity of similar data and increased complexity arising from intricate object compositions. Our method builds on the insight of repurposing the diffusion model as a novel view synthesizer while emphasizing the inherent properties of multi-object scenarios in both model design and training strategy to facilitate multi-object NVS.

### 3.2 MOVIS

Our proposed method extends view-conditioned diffusion models to multi-object level, as illustrated in Fig. 2. The model leverages a pre-trained Stable Diffusion (Rombach et al., 2022) and concatenates the 2D structural information from the input view with a noisy target image as input. Additionally, it integrates a pre-trained image encoder (Oquab et al., 2023) to capture semantic information, which is injected into the network through cross-attention alongside the relative camera pose. Moreover, it predicts novel view mask simultaneously as an auxiliary task to aid global object placement learning.

**Structure-Aware Feature Amalgamation**  To synthesize plausible images under novel viewpoints, the model must first grasp the compositional structural information from the input view, laying a solid foundation for generation. To address the innate complexity in multi-object scenarios due to the intricate object relationship, we propose to leverage structure-aware features to facilitate model's comprehension. Specifically, we use depth maps and object masks as proxies for image-level structural information. Object masks provide a rough concept of object placement and shape as well as distinguishing distinct object instances, while depth maps encode the rough relative position and shape of the visible objects. Together with input-view images, these conditions provide both global structural information like object placement and local fine-grained details like object shape. Concretely speaking, we normalize the image rendered with object instance IDs of the input view to create a continuous object mask image $\widehat{\mathbf{M}}$. We then replicate the depth map $\widehat{\mathbf{D}}$ and object mask image $\widehat{\mathbf{M}}$ into three channels to simulate RGB images. These two structural-aware feature images, along with the input image $\hat{\mathbf{x}}_0$, are passed into a VAE to obtain latent features, which will be later concatenated with the noisy target view image $\mathbf{x}_t$ as input to the denoising U-Net. Note that both object mask and depth can be obtained with off-the-shelf detectors during the inference stage, such as SAM (Kirillov et al., 2023) and Marigold (Ke et al., 2024). After introducing these additional conditions, the learning objective of MOVIS becomes:

$$\mathbb{E}[||\epsilon_\theta(\alpha_t\mathbf{x}_0 + \sigma_t\epsilon, t, C_{\text{SA}}(\hat{\mathbf{x}}_0, R, T, \widehat{\mathbf{D}}, \widehat{\mathbf{M}})) - \epsilon||_2^2]. \quad (3)$$

We use $C_{\text{SA}}(\cdot)$ as a shorthand for the structure-aware view-conditioned feature throughout the paper.

**Auxiliary Novel View Mask Prediction Task**  Input-view depth maps and mask images are intended to help the model indirectly understand the structure of multi-object compositions by incorporating additional structure-aware information into the input. To encourage the model to better grasp overall structure, particularly its ability to generate it, we propose leveraging structural information (*i.e.*, mask image) prediction under the target view as an auxiliary task, providing more direct supervision. Our approach draws inspiration from classifier guidance (Dhariwal & Nichol, 2021), where a classifier $p_\phi(y|x_t, t)$ guides the denoising process of image $x_t$ to meet the criterion $y$ via incorporating the gradient $\nabla_{x_t}\log p_\phi((y|x_t, t))$ during the inference process as an auxiliary guidance. Similarly, to improve the model's ability to learn compositional structure, particularly in synthesizing novel view plausible object placement (position and orientation), we introduce an auxiliary task during training: predicting object mask images $\mathbf{M}_t \sim p(\mathbf{M}_t|\mathbf{x}_t, t, C_{\text{SA}}(\cdot))$ under target view. This prediction is conditioned on the noisy target-view image $\mathbf{x}_t$, timestep $t$ and input-view structure-aware feature $C_{\text{SA}}(\cdot)$, derived from the final layer of the denoising U-Net. The supervision could be formulated as:

$$\nabla_{\mathbf{x}_t}\log p(\mathbf{x}_t, \mathbf{M}_t|t, C_{\text{SA}}(\cdot)) = \nabla_{\mathbf{x}_t}\log p(\mathbf{x}_t|t, C_{\text{SA}}(\cdot)) + \nabla_{\mathbf{x}_t}\log p(\mathbf{M}_t|\mathbf{x}_t, t, C_{\text{SA}}(\cdot)). \quad (4)$$

Following Eq. (4), we jointly train the mask predictor and denoising U-Net following:

$$\mathbb{E}[||\epsilon_\theta(\alpha_t\mathbf{x}_0 + \sigma_t\epsilon, t, C_{\text{SA}}(\cdot)) - \epsilon||_2^2 + \gamma||\mathbf{M}_{tgt} - \mathbf{M}_t||_2^2], \quad (5)$$

where we use $\mathbf{M}_{tgt}$ to denote the ground-truth target-view image, and we use the weight $\gamma = 0.1$ to balance the diffusion loss and mask prediction loss.

## 3.3 STRUCTURE-GUIDED TIMESTEP SAMPLING SCHEDULER

Inspired by previous works (Jiang et al., 2023; Chen, 2023) that identify the importance of different scheduling strategies, we first provide an in-depth analysis of the inference process of multi-object novel view synthesis, where we adopt a DDIM (Song et al., 2020a) sampler:

$$\mathbf{x}_{t-1} = \sqrt{\alpha_{t-1}}\underbrace{\left(\frac{\mathbf{x}_t - \sqrt{1-\alpha_t} \cdot \mathbf{F}}{\sqrt{\alpha_t}}\right)}_{\text{predicted } \mathbf{x}_0} + \sqrt{1 - \alpha_{t-1} - \sigma_t^2} \cdot \mathbf{F} + \sigma_t\epsilon_t. \quad (6)$$

We use $\mathbf{F}$ as a shorthand for $\epsilon_\theta(\mathbf{x}_t, t, C_{\text{SA}}(\cdot))$ and $\epsilon_t \sim \mathcal{N}(\mathbf{0}, \mathbf{I})$. We examine the predicted $\mathbf{x}_0$ (as in Eq. (6)) and the predicted mask image $\mathbf{M}_t$ at various timesteps during the inference process as they offer direct visualizations for analysis. These visualized results are presented in Fig. 3.

In Fig. 3, we observe that a blurry image, which indicates the approximate placement of each object, is quickly restored in the early stages (*i.e.*, larger $t$) of the inference process. This suggests

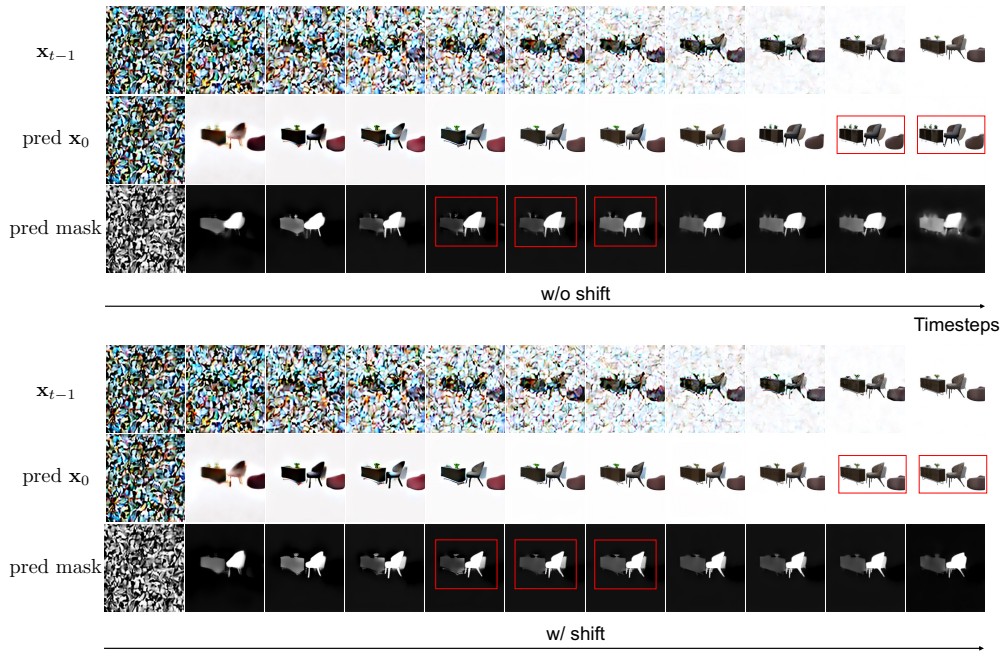

Figure 3: **Visualization of inference.** The early stage of the denoising process focuses on restoring global object placements, while the prediction of object masks requires a relatively noiseless image to recover fine-grained geometry. This motivates us to seek a balanced timestep sampling scheduler during training. The model trained *w/ shift* yields better mask prediction and thus recovers an image with more details and sharp object boundary. The *w/o shift* here refers to not shifting the $\mu$ value.

that global structural information is prioritized for the model to learn during this stage. Accurate object placements are crucial for synthesizing reasonable novel view images, as incorrect placement predictions indicate a fundamental misunderstanding of the compositional structure. This underscores the importance of training the model with a larger $t$ during the initial training periods, which is even more important for multi-object NVS scenarios considering the increased compositional complexity compared with a single object. Conversely, a mask with a clear boundary is not predicted until a later stage of the sampling process (*i.e.*, smaller $t$). This is because accurate mask prediction depends heavily on a relatively noiseless image. Therefore, to capture fine-grained geometry and appearance details of objects, it is essential to train the model with a smaller $t$ during later training periods.

Recognizing the importance of timestep $t$ in balancing the learning of global placement information and local fine-grained details, we propose to adjust the original timestep sampling process to:

$$t \sim \mathcal{U}(1, 1000) \to t \sim \mathcal{N}(\mu(s), \sigma), \text{ where } \mu(s) = \mu_{\text{local}} + (\mu_{\text{global}} - \mu_{\text{local}}) \cdot \frac{s}{T_s} \tag{7}$$

where $s$ denotes the model training iteration, $T_s$ denotes the total number of training steps, $\sigma = 200$ is a constant variance. We sample the timestep $t$ from a Gaussian distribution with mean $\mu(s)$ following a linear decay from a large value $\mu_{\text{global}} = 1000$ to a small value $\mu_{\text{local}} = 500$. This approach allows the model to initially learn correct global object placement information and gradually turn its focus to refining detailed object geometry in later training stages. In practice, we include a warmup period with 4000 training steps sampling $t$ with a fixed $\mu(s) = \mu_{\text{global}}$. After the warmup, we use the linear decay schedule over 2000 steps, and then stabilize the learning for fine-grained details after 6000 steps where we use $\mu(s) = \mu_{\text{local}}$.

## 4 EXPERIMENTS

### 4.1 EXPERIMENT SETUP

We focus on multi-object composite NVS tasks in indoor scenes, with an emphasis on foreground objects, examining novel view structural plausibility regarding object placement, geometry, appear-

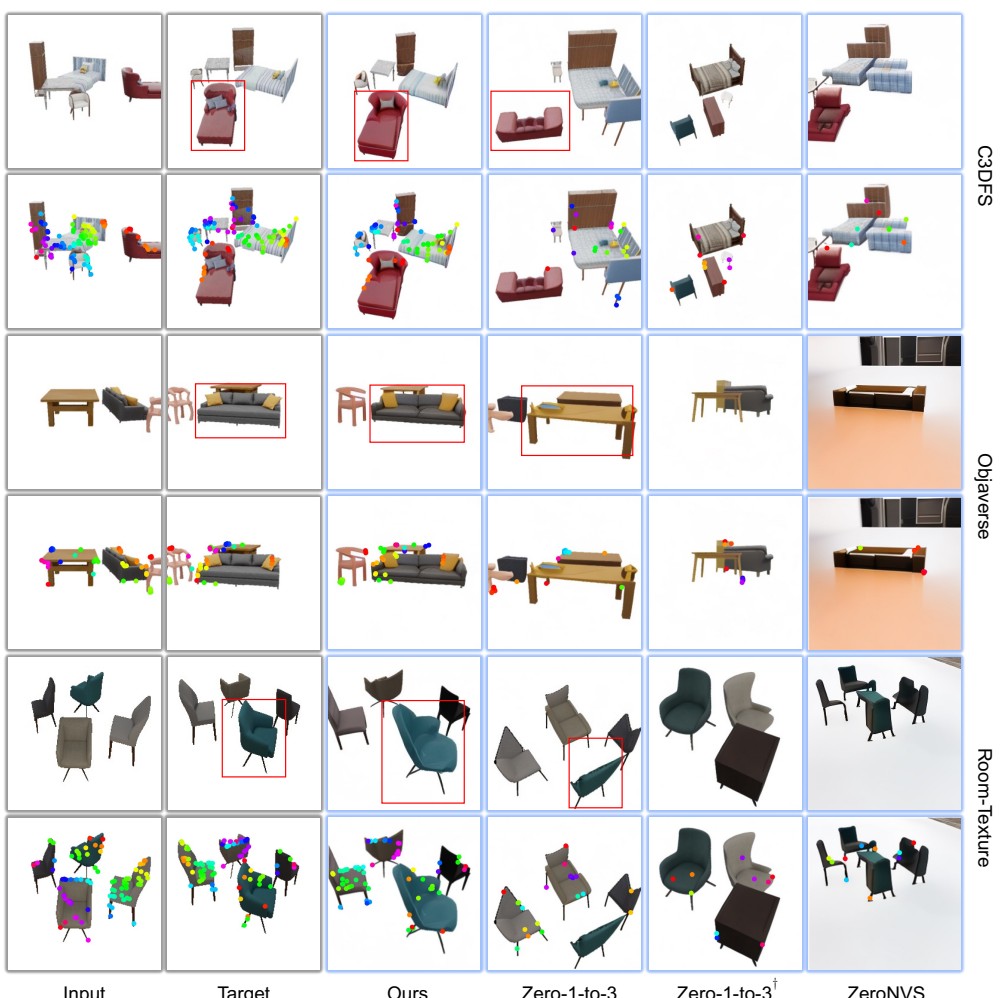

Figure 4: **Qualitative results of NVS and cross-view matching.** Our method generates plausible novel-view images across various datasets, surpassing baselines regarding object placement, shape, and appearance. In cross-view matching, points of the same color indicate correspondences between the input and target views. We achieve a higher number of matched points with more precise locations.

ance, and cross-view consistency with input view. This choice stems from the recent advancements in object segmentation (Kirillov et al., 2023), while we leave the background modeling for future work.

**Datasets.** To facilitate the training and evaluation of our proposed method, we curate a scalable synthetic dataset Compositional 3D-FUTURE (C3DF), comprising 100k composites for training and 5k for testing. Each composite is created by composing pre-filtered furniture items from 3D-FUTURE (Fu et al., 2021b) using a heuristic strategy to avoid collision and penetration. Beyond C3DF, we emphasize testing the generalization capability by benchmarking our method on Room-Texture (Luo et al., 2024) and Objaverse (Deitke et al., 2023b). We also evaluate our model on diverse indoor scenes from both the synthetic dataset 3D-FRONT (Fu et al., 2021a) and the real-world dataset SUNRGB-D (Song et al., 2015). Refer to Appx. B.2 for more dataset details.

**Baselines.** We compare our method against two recent novel view synthesis methods including Zero-1-to-3 (Liu et al., 2023c) and ZeroNVS (Sargent et al., 2023). The original Zero-1-to-3 is trained on extensive object-level datasets. Therefore, we also re-train Zero-1-to-3 on our synthetic dataset C3DF, denoted as Zero-1-to-3[†]. ZeroNVS is trained on a mixture of real-world images with background, so we use images with backgrounds as its input if possible for a fair comparison.

Table 1: **Quantitative results of multi-object NVS, Object Placement, and Cross-view Consistency**. We evaluate on C3DF test set, along with *generalization experiments* on Room-Texture (Luo et al., 2024) and Objaverse (Deitke et al., 2023b). [†] indicates re-training on C3DF.

| Dataset | Method | Novel View Synthesis | | | Placement | Cross-view Consistency | |
| | | PSNR(↑) | SSIM(↑) | LPIPS(↓) | IoU(↑) | Hit Rate(↑) | Dist(↓) |
| --- | --- | --- | --- | --- | --- | --- | --- |
| C3DF | ZeroNVS | 10.704 | 0.533 | 0.481 | 21.6 | 1.2 | 130.3 |
| | Zero-1-to-3 | 14.255 | 0.771 | 0.302 | 33.7 | 5.8 | 86.9 |
| | Zero-1-to-3[†] | 14.811 | 0.794 | 0.283 | 34.4 | 1.6 | 120.3 |
| | Ours | **17.432** | **0.825** | **0.171** | **58.1** | **37.0** | **44.8** |
| Room-Texture | ZeroNVS | 8.217 | 0.647 | 0.487 | 8.2 | 1.2 | 140.3 |
| | Zero-1-to-3 | 9.860 | 0.712 | 0.406 | 13.9 | 2.9 | 104.1 |
| | Zero-1-to-3[†] | 8.342 | 0.657 | 0.452 | 13.5 | 0.5 | 157.4 |
| | Ours | **10.014** | **0.718** | **0.366** | **24.2** | **6.1** | **78.1** |
| Objaverse | ZeroNVS | 10.557 | 0.513 | 0.486 | 17.3 | 2.3 | 126.9 |
| | Zero-1-to-3 | 15.850 | 0.810 | 0.259 | 34.7 | 10.7 | 80.7 |
| | Zero-1-to-3[†] | 15.433 | 0.815 | 0.273 | 29.7 | 1.7 | 126.7 |
| | Ours | **17.749** | **0.840** | **0.169** | **51.3** | **50.0** | **47.2** |

**Metrics.** We utilize PSNR, SSIM, and LPIPS as metrics for evaluating the quality of *Novel View Synthesis*. To assess global object *Placement*, we compute the foreground-background IoU with ground-truth masks. Finally, we propose metrics to evaluate *Cross-view Consistency* with image-matching. More specifically, we first apply MASt3R (Leroy et al., 2024) to acquire the image matching between the input-view image and target-view image for both ground truth and model predictions. With the ground-truth matching as references, we compute each method's *Hit Rate* and the nearest matching distance (*Dist.*). *Hit Rate* measures the proportion of predicted matches that align with the ground truth matches. *Dist.* quantifies the distance between the predicted matching and ground-truth matching in the target view. Please refer to Appx. B.3 for more details about the metrics.

## 4.2 RESULTS AND DISCUSSIONS

Fig. 4 presents qualitative results of multi-object NVS and cross-view matching visualization on different datasets, with quantitative results in Tab. 1. We summarize the following key observations:

1. Our method realizes the highest PSNR and generates high-quality images under novel views, closely aligned with the ground truth images, especially regarding novel-view object placement (position and orientation), shape, and appearance. In contrast, the baseline models struggle to accurately capture the compositional structure under novel views. For example, in the first row, the red bed is incorrectly oriented in Zero-1-to-3 and is either missing or distorted in other baselines.

2. From the visualized cross-view matching results and the metrics in Tab. 1, it is evident that our method significantly outperforms the baseline approaches in *Cross-view Consistency*. It achieves a much higher IoU and *Hit Rate* while exhibiting a considerably lower matching distance. The visualized results are consistent with the metrics, further validating our method's accuracy in capturing cross-view structural consistency, which cannot be reflected by existing NVS metrics.

3. Our model exhibits strong generalization capabilities on unseen datasets, *e.g.*, Room-Texture and Objaverse. We demonstrate more qualitative results, including on 3D-FRONT and SUNRGB-D, in Appx. B.4. We showcase potential applications, including object removal and reconstruction in Appx. B.5. Further discussion about limitation and failure cases are presented in Appx. C.

Table 2: **Ablation results on C3DF**.

| Dataset | Method | Novel View Synthesis | | | Placement | Cross-view Consistency | |
| | | PSNR(↑) | SSIM(↑) | LPIPS(↓) | IoU(↑) | Hit Rate(↑) | Dist(↓) |
| --- | --- | --- | --- | --- | --- | --- | --- |
| C3DF | w/o depth | 17.080 | 0.819 | 0.178 | 57.2 | **39.2** | 45.2 |
| | w/o mask | 16.914 | 0.818 | 0.187 | 54.7 | 25.4 | 50.4 |
| | w/o sch. | 16.166 | 0.808 | 0.212 | 49.1 | 11.9 | 48.6 |
| | Ours | **17.432** | **0.825** | **0.171** | **58.1** | 37.0 | **44.8** |

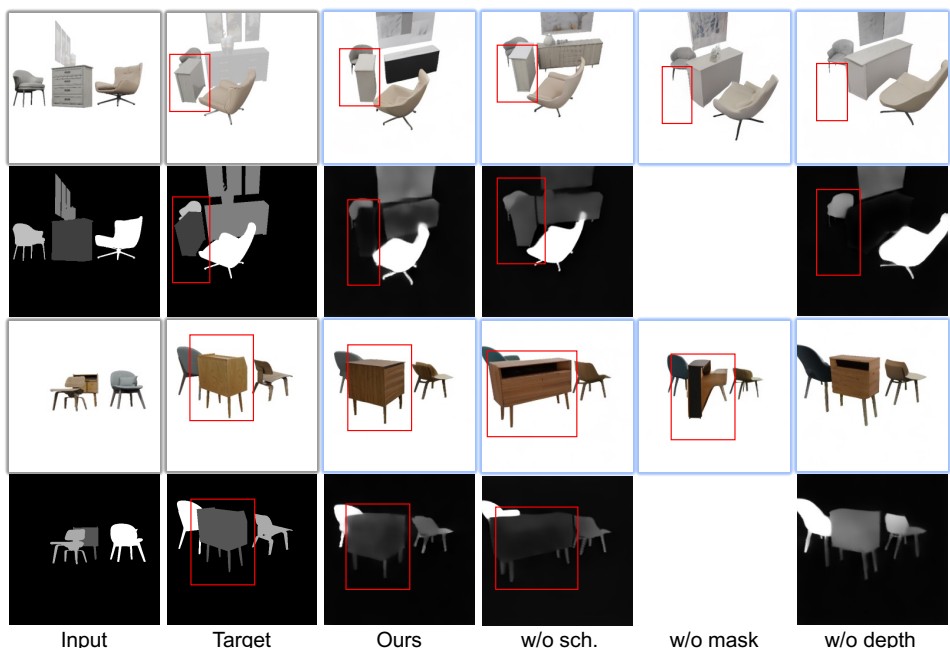

Figure 5: **Qualitative comparison for ablation study**. Removing depth or mask predictions weakens the model's understanding of object placement and existence, exemplified by the missing white cabinet in the first example. Excluding mask predictions or the scheduler reduces the model's ability to learn object placement, as shown by the misoriented brown cabinet in the second example.

### 4.3 ABLATION STUDY

To verify the efficacy of each component, we perform an ablation study on our key technical designs, including the depth input (*w/o depth*), mask prediction auxiliary task (*w/o mask*), and the scheduler (*w/o sch.* learns with a uniform sampler $t \sim \mathcal{U}(1, 1000)$ ). Results in Tab. 2 show that the auxiliary mask prediction task and the timestep sampler are the most critical components, significantly affecting all the metrics and the realistic object recovery as demonstrated by the misoriented brown cabinet in the second example from Fig. 5. Without the scheduler, the model produces less accurate object positions, evident both qualitatively and quantitatively. Furthermore, removing depth or mask predictions weakens the model's understanding of spatial relationships and object existence, exemplified by the completely missing white cabinet in the first example. This also shows incorporating structure-aware features as inputs, though seemingly intuitive, offers the most straightforward approach to enhancing the model's structural awareness, particularly given recent advancements in monocular predictors (Kirillov et al., 2023; Ke et al., 2024). We present a more comprehensive discussion on the scheduler strategy in Appx. A.4 and ablations on more datasets are in Appx. B.4.

## 5 CONCLUSION

We extend diffusion-based novel view synthesizers to handle multi-object compositions in indoor scenes. Our proposed model generalize well across diverse datasets with more accurate object placement, shape, and appearance, showing a stronger cross-view consistency with input view. The core of our approach lies in integrating structure-aware features as additional inputs, an auxiliary mask prediction task, and a structure-guided timestep sampling scheduler. These components enhance the model's awareness of compositional structure while balancing the learning of global object placement and fine-grained local shape and appearance. Given the prevalence of multi-object compositions in real-world scenes, we believe that our model designs and comprehensive evaluations can offer valuable insights for advancing scene-level NVS models in more complex environments.

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

## A  MODEL DETAILS

### A.1  DINO PATCH FEATURE AND CAMERA VIEW EMBEDDING

The original image encoder of Stable Diffusion is CLIP, which excels at aligning images with text. Other image encoders like DINO-v2 (Oquab et al., 2023) or ConvNeXtv2 (Woo et al., 2023) may provide denser image features that may benefit generation tasks as mentioned by previous works (Jiang et al., 2023; Kong et al., 2024). Therefore, we opt to use the DINO feature instead of the original CLIP feature in our network following Jiang et al. (2023). To inject the DINO patch feature into our network, we encode the input view image using DINO-v2 (Oquab et al., 2023) "norm patchtokens", whose shape dimension is $(b, 16, 16, 1024)$. We will simply flatten it into $(b, 256, 1024)$ to apply cross-attention, and $b$ means batch size here.

As for the camera view embedding, we choose to embed it using a 6 degrees of freedom (6DoF) representation. To be specific, let $E_i$ be the extrinsic matrix under the input view and $E_j$ be the extrinsic matrix under the output view, we represent relative camera pose change as $E_i^{-1} E_j$. We will also flatten it into 16 dimensions to concatenate it to the image feature. Afterwards, we will replicate the 16-dimension embedding 256 times to concatenate the embedding to every channel of the DINO feature map. A projection layer will later be employed to project the feature map into $(b, 256, 768)$ to match the dimension of the CLIP encoder, which was originally used by Stable Diffusion so that we can fine-tune the pre-trained checkpoint. It is worth noting that we also tried other novel view synthesizer's camera embedding like Zero-1-to-3 (Liu et al., 2023c) using a 3DoF spherical coordinates in early experiments, but we found that it does not make much of a difference.

### A.2  DEPTH AND MASK CONDITION

In this section, we will explain how input view depth and mask are incorporated as additional conditioning inputs. For depth maps, regions with infinite depth values are assigned a value equal to twice the maximum finite depth value in the rest of the image. After this adjustment, we apply a normalization technique to scale the depth values to the range of $[-1, 1]$, enabling the use of the same VAE architecture as for images.

For mask images, we assign unique values to different object instances in the input view. For instance, if there are four objects in the multi-object composite, they will be labeled as 1, 2, 3, and 4, respectively, while the background will be assigned a value of 0. The same normalization technique used for depth maps is applied to these mask images. These mask images, like all other inputs, are processed by the VAE, with all images set to a resolution of $256 \times 256$.

### A.3  SUPERVISION FOR AUXILIARY MASK PREDICTION TASK

To implement the auxiliary mask prediction task, we encode the output view mask images into the same latent space as the input view mask images. Object instances viewed from different angles will be assigned the same value, which is ensured during the curation of our compositional dataset. Supervision is directly applied to the latent mask features extracted from the final layer of the denoising U-Net. Only the input view mask images are required during inference, simplifying the process while preserving consistency across views.

### A.4  TIMESTEP SCHEDULER

Though we finally employed a linearly declining strategy, we experimented with several alternatives. Specifically, we tested linearly declining the mean of the Gaussian distribution (LDC), linearly increasing the mean after a sudden drop (LIND), and keeping the mean constant (KMS). These strategies are illustrated in Fig. A1. The metrics on the test set of our C3DF are provided in Tab. A1, with some visual comparisons in Fig. A2. *w/o sch.* in Tab. A1 refers to applying a uniform sampler, same as the one in the main paper. From the results, we observe that LDC achieves slightly better performance than LIND and KMS, largely outperforming *w/o sch.*

However, we observed significant visual artifacts such as weird colors and extremely blurry mask images when combining the auxiliary mask prediction task with the KMS sampling strategy, as demonstrated in Fig. A2. For example, the bed in the second example possesses unclear object

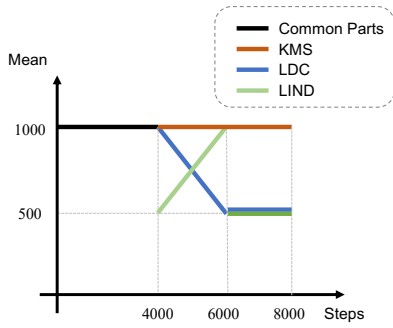

Figure A1: **Illustration of different timestep sampling strategies.**

Table A1: **Ablation on different strategies.** Incorporating sampling strategies significantly improves the model performance, while the linear decline (LDC) achieves the best.

| Dataset | Method | Novel View Synthesis | | |
|---------|--------|------------|-----------|------------|
| | | PSNR(↑) | SSIM(↑) | LPIPS(↓) |
| C3DF | w/o sch. | 16.166 | 0.808 | 0.212 |
| | KMS | 17.148 | 0.823 | 0.175 |
| | LIND | 17.279 | 0.824 | 0.172 |
| | LDC | **17.432** | **0.825** | **0.171** |

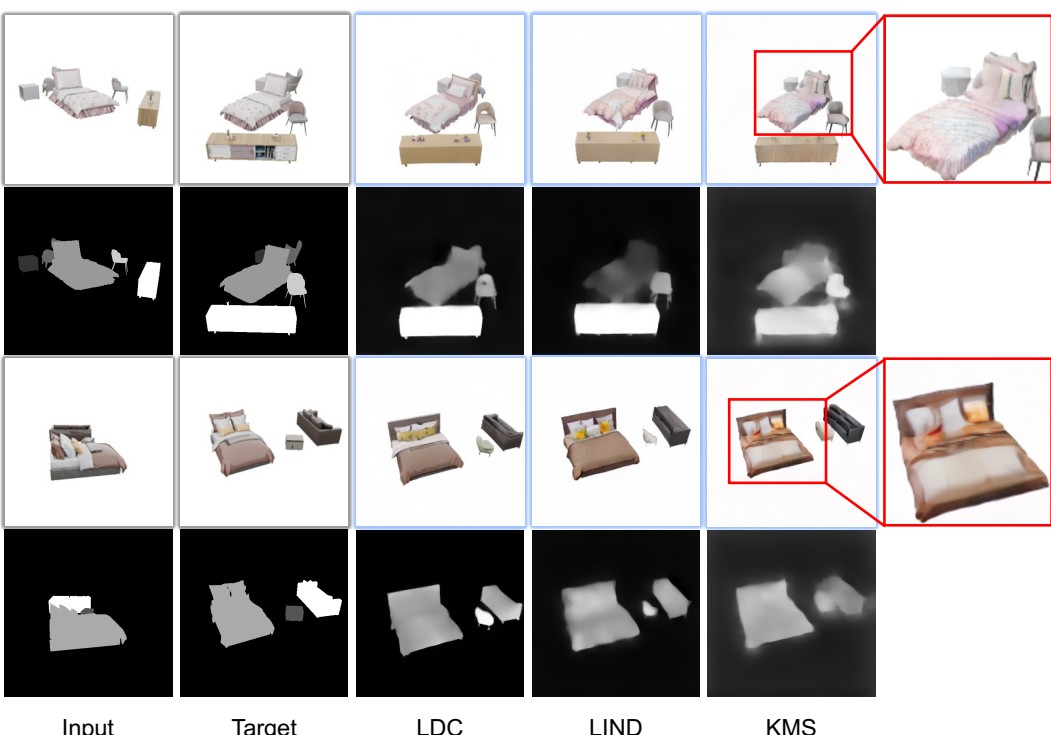

Figure A2: **Comparison of different strategies**. The predicted images and mask images under novel views using different strategies are visualized. We can observe that images predicted by the KMS strategy possess weird and blurry color while LDC strategy seems to be slightly better than LIND.

boundaries and distorted texture. We believe this is due to KMS focusing primarily on denoising at larger timesteps, which provides limited guidance for recovering mask images and refining fine-grained geometry and appearance. Consequently, without a dedicated period for denoising smaller timesteps, the per-object shape and appearance appear distorted and unrealistic.

## B EXPERIMENT DETAILS

### B.1 IMPLEMENTATION DETAILS

We solely utilize the data from C3DF as the training source for our model. The training process takes around 2 days on 8 NVIDIA A100 (80G) GPUs, employing a batch size of 172 per GPU. The exact training steps are 8,000 steps. During the inference process, we apply 50 DDIM steps and set the

guidance scale to 3.0. We use DepthFM (Gui et al., 2024) and SAM (Kirillov et al., 2023) to extract the depth maps and object masks when they are not available, as well as for all real-world images.

## B.2 DATASETS

**C3DF**   We use the furniture models from the 3D-FUTURE dataset (Fu et al., 2021b) to create our synthetic multi-object compositional data. The 3D-FUTURE dataset contains 9,992 detailed 3D furniture models with high-resolution textures and labels. Following previous work Chen et al. (2024a), we categorize the furniture into seven groups: bed, bookshelf, cabinet, chair, nightstand, sofa, and table. To ensure unbiased evaluation, we further split the furniture into distinct training and test sets, ensuring that none of the test set items are seen during training.

After filtering the furniture, we first determine the number of pieces to include in each composite, which is randomly selected to be between 3 and 6. Next, we establish a probability distribution based on the different types of furniture items and sample each piece according to this distribution. To prevent collisions and penetration between furniture items, we employ a heuristic strategy. Specifically, for each furniture item to be added, we apply a random scale adjustment within the range of $[0.95, 1.05]$, as the inherent scale of the furniture models accurately reflects real-world sizes. We also rotate each model by a random angle to introduce additional variability. Once these adjustments are complete, we begin placing the furniture items in the scene. The first item is positioned at the center of the scene at coordinates $(0, 0, 0)$. Subsequent objects are added one by one, initially placed at the same central location. Since this results in inevitable collisions, we randomly sample a direction and gradually move the newly added item along this vector until there is no intersection between the bounding boxes of the objects. By following these steps, we generate a substantial number of multiple furniture items composites, ultimately creating a training set of 100,000 composites and a test set of 5,000 to evaluate the capabilities of our network.

After placing all the furniture items, we render multi-view images to facilitate training, using Blender (Community, 2018) as our renderer due to its high-quality output. We first normalize each composite along its longest axis. To simulate real-world camera poses and capture meaningful multi-object compositions, we employ the following method for sampling camera views.

Cameras are randomly sampled using spherical coordinates, with a radius range of $[1.3, 1.7]$ and an elevation angle range of $[2°, 40°]$. There are no constraints on the azimuth angle, allowing the camera to rotate freely around multiple objects. The chosen ranges for the radius and elevation angles are empirical. In addition to determining the camera positions, we establish a "look-at" point to compute the camera pose. This point is randomly selected on a spherical shell with a radius range of $[0.01, 0.2]$.

To enhance the model's compositional structural awareness, we also render depth maps and instance masks (both occluded and unoccluded) from 12 different viewpoints. The unoccluded instance mask ensures that if one object is blocked by another, the complete amodal mask of the occluded object is still provided, regardless of any obstructions. Although these unoccluded instance masks are not currently necessary for our network, we render them for potential future use.

**Objaverse**   To evaluate our network's generalization capability, we create a small dataset comprising 300 composites sourced from Objaverse (Deitke et al., 2023b). Specifically, we utilize the provided LVIS annotations to select categories that are commonly found in indoor environments, such as beds, chairs, sofas, dressers, tables, and others. Since the meshes from Objaverse vary in scale, we rescale each object based on reference object scales from the 3D-FUTURE dataset (Fu et al., 2021b). The composition and rendering processes follow the same strategy employed in C3DF.

Table A2: **Availability of conditions.**

|       | C3DFS | Room-Texture | Objaverse | SUNRGB-D | 3D-FRONT |
|-------|-------|--------------|-----------|----------|----------|
| depth | ✓     | ×            | ✓         | ×        | ×        |
| mask  | ✓     | ✓            | ✓         | ×        | ×        |

**Inference Details**  Since our model requires input-view depth map and mask images as additional inputs, we need to use DepthFM (Gui et al., 2024) and SAM (Kirillov et al., 2023) to extract the depth maps and object masks when they are not available, as well as for all real-world images. We show whether all the used datasets have provided depth maps and mask images in Tab. A2. '×' means they do not provide such conditions while '✓' means they do provide such conditions.

### B.3  METRICS

**Intersection over Union (IoU)**  Since all baseline methods do not possess the concept of every object instance, we compute a foreground-background IoU for comparison. This metric can provide a rough concept of the overall placement alignment with ground truth images. We extract the foreground object mask by converting the generated image to grayscale ($I_L$). Given that the generated image has a white background, we compute the foreground mask $\mathbf{M}$ as $\mathbf{M} = I_L < \beta_{th}$, where $\beta_{th}$ is a threshold that is fixed as 250.

**Cross-view Matching**  As outlined in the main paper, we introduce two metrics to systematically evaluate cross-view consistency with the input view: **Hit Rate** and **Nearest Matching Distance**. Since direct assessment of cross-view consistency is not feasible by merely evaluating the success matches between each method's predicted novel view images and the input view image, we opt to how far the predicted matches deviate from the ground-truth matches.

We first compute ground-truth matching points and every model's matching points using MASt3R (Leroy et al., 2024) upon the input view image and the output view image (ground truth or predicted). Each matching pair is represented as a four-element tuple $(\mathbf{x}^0, \mathbf{y}^0, \mathbf{x}^1, \mathbf{y}^1)$, where $(\mathbf{x}^0, \mathbf{y}^0)$ corresponds to the point on the input-view image, and $(\mathbf{x}^1, \mathbf{y}^1)$ corresponds to the point on the output-view image.

For each ground-truth matching pair $(\mathbf{x}_{gt}^0, \mathbf{y}_{gt}^0, \mathbf{x}_{gt}^1, \mathbf{y}_{gt}^1)$, we find the nearest predicted matching pair in each model's results, denoted as $(\mathbf{x}^0, \mathbf{y}^0, \mathbf{x}^1, \mathbf{y}^1)$, based on the Euclidean distance between points in the input view image. If both $\mathbf{L}_2||(\mathbf{x}_{gt}^0, \mathbf{y}_{gt}^0), (\mathbf{x}^0, \mathbf{y}^0)||$ and $\mathbf{L}_2||(\mathbf{x}_{gt}^1, \mathbf{y}_{gt}^1), (\mathbf{x}^1, \mathbf{y}^1)||$ is smaller than a fixed threshold 20, the match is considered a successful hit. The **Hit Rate** is then calculated as the ratio of successful hits to the total number of ground-truth matches.

For Nearest Matching Distance, we examine whether $\mathbf{L}_2||(\mathbf{x}_{gt}^0, \mathbf{y}_{gt}^0), (\mathbf{x}^0, \mathbf{y}^0)||$ is within the threshold. For those passing this check, we compute the mean distance $\mathbf{L}_2||(\mathbf{x}_{gt}^1, \mathbf{y}_{gt}^1), (\mathbf{x}^1, \mathbf{y}^1)||$ as the **Nearest Matching Distance**, averaging over all successful hits. A detailed pseudo-code explanation can be found in Alg. 1 and Alg. 2.

---

**Algorithm 1** Hit Rate Computation

---

1: // Obtain image-matching pairs using MASt3R and save in a list
2: $\text{Pairs}_{gt} = \text{MASt3R(GT)}$
3: $\text{Pairs}_{ours} = \text{MASt3R(Ours)}$
4: // Each element in the list is a four-element tuple $\text{p} = (\mathbf{x}^0, \mathbf{y}^0, \mathbf{x}^1, \mathbf{y}^1)$
5: // $(\mathbf{x}^0, \mathbf{y}^0)$ refers to the point in the input view image and $(\mathbf{x}^1, \mathbf{y}^1)$ the point in output view image
6: $\text{Hits} = 0$
7: **For** $\text{p}_{gt}$ **in** $\text{Pairs}_{gt}$
8:     // $\text{p}_{ours}^i$ is the $i$-th element of $\text{Pairs}_{ours}$
9:     // p[:2] refers to the first two element in the tuple and p[2:] the last two
10:     $i^\star = \arg\min_i(\mathbf{L}_2(\text{p}_{gt}[:2], \text{p}_{ours}^i[:2]))$
11:     **IF** $\mathbf{L}_2(\text{p}_{gt}[:2], \text{p}_{ours}^{i^\star}[:2]) < 20$ **and** $\mathbf{L}_2(\text{p}_{gt}[2:], \text{p}_{ours}^{i^\star}[2:]) < 20$
12:         // Successfully hit one, delete it from gt pairs and ours pairs
13:         $\text{Hits} \leftarrow \text{Hits} + 1$
14:         **POP**$(\text{Pairs}_{ours}, \text{p}_{ours}^{i^\star})$
15: return $\text{Hits}/\text{len}(\text{Pairs}_{gt})$

---

---

**Algorithm 2** Nearest Matching Distance Computation

---

1: // The notations are the same as the one in Alg. 1
2: $\text{Pairs}_{\text{gt}} = \text{MASt3R}(\textbf{GT})$
3: $\text{Pairs}_{\text{ours}} = \text{MASt3R}(\textbf{Ours})$
4: $\text{Dist} = \text{EmptyList}()$
5: **For** $\text{p}_{\text{gt}}$ **in** $\text{Pairs}_{\text{gt}}$
6:      $i^{\star} = \arg\min_{i}(\mathbf{L}_2(\text{p}_{\text{gt}}[:2], \text{p}_{\text{ours}}^{i}[:2]))$
7:      **IF** $\mathbf{L}_2(\text{p}_{\text{gt}}[:2], \text{p}_{\text{ours}}^{i^{\star}}[:2]) < 20$
8:          **Append**$(\text{Dist}, \mathbf{L}_2(\text{p}_{\text{gt}}[2:], \text{p}_{\text{ours}}^{i^{\star}}[2:]))$
9:          **POP**$(\text{Pairs}_{\text{ours}}, \text{p}_{\text{ours}}^{i^{\star}})$
10: return $\text{Mean}(\text{Dist})$

---

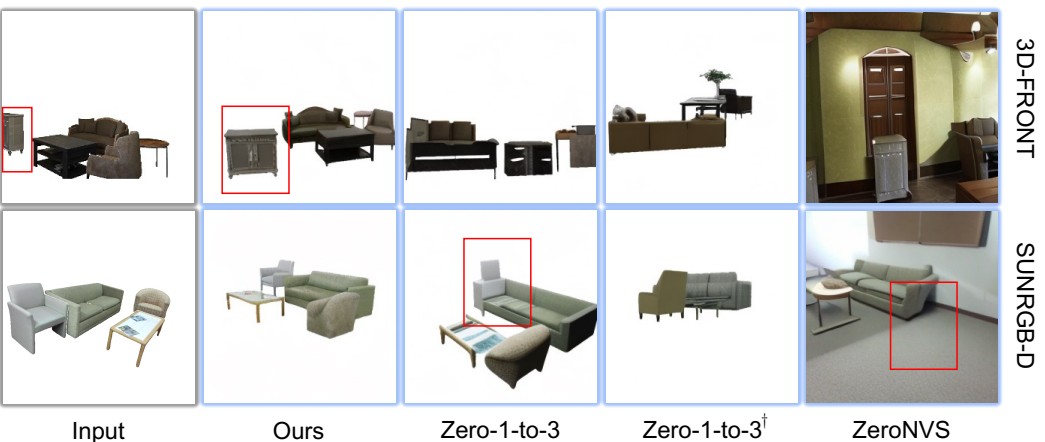

| Input | Ours | Zero-1-to-3 | Zero-1-to-3$^{\dagger}$ | ZeroNVS |
|---|---|---|---|---|

Figure A3: **Visualized comparison with baselines**. Our method synthesizes more consistent novel view images and can even hallucinate objects that exceed the edge of image as shown in the first row. Conversely, baselines may predict unclear object boundary and omit objects under novel views.

### B.4 RESULTS

We show more visualized results of our own methods along with ground truth on C3DF in Fig. A11, on Objaverse (Deitke et al., 2023b) in Fig. A12, and on Room-Texture (Luo et al., 2024) in Fig. A13. More visualized comparisons with baselines on SUNRGB-D (Song et al., 2015) and 3D-FRONT Fu et al. (2021a) are shown in Fig. A3. A more complete ablation study on other datasets including Objaverse and Room-Texture is shown in Tab. A3. Some continuous rotation examples on SUNRGB-D are shown in Fig. A4, on 3D-FRONT are shown in Fig. A5, and more cross-view matching results without ground-truth pairs as reference are shown in Fig. A6.

### B.5 APPLICATIONS

**Object Removal**    Since we can predict mask images under novel views, we can support simple image editing tasks like novel view object removal by simply setting a threshold value in the mask image and mask out corresponding pixels to achieve object removal. An example is shown in Fig. A8.

**Reconstruction**    The capability to synthesize novel view images that are consistent with the input view image demonstrates that the model possesses 3D-awareness, which can assist downstream tasks such as reconstruction. We leverage an off-the-shelf reconstruction method DUSt3R (Wang et al., 2024) using the input-view image and novel view images predicted by our method. Two visualized examples are shown in Fig. A9.

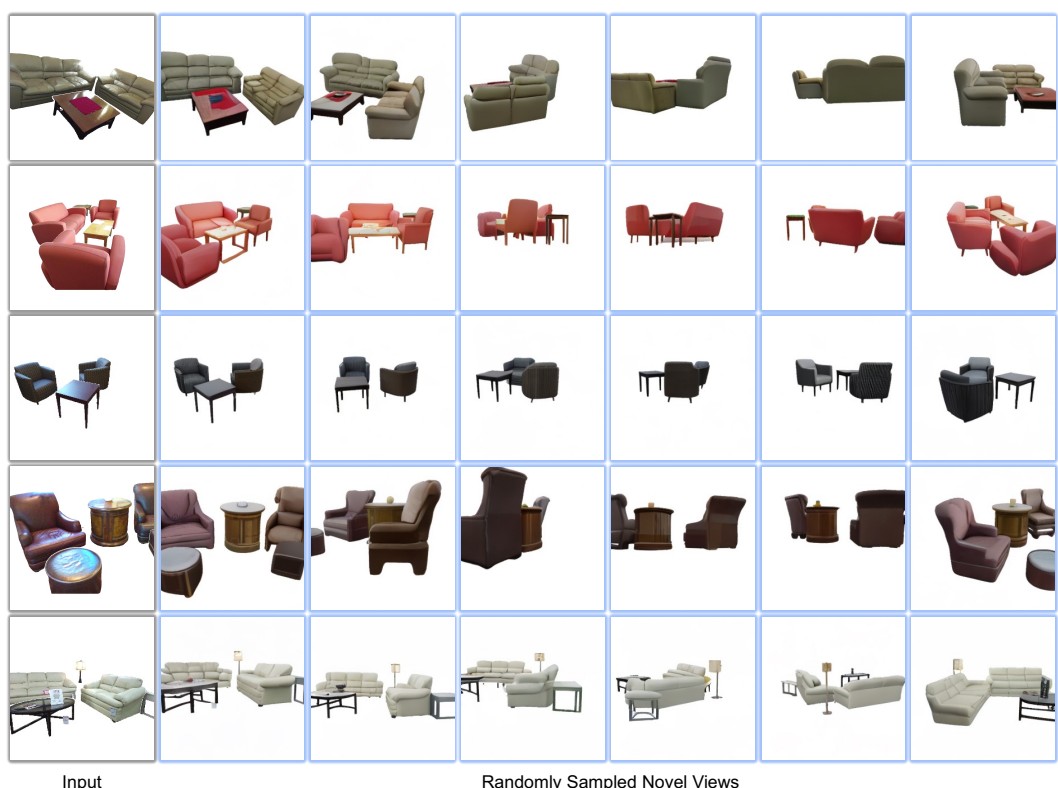

Input                                                    Randomly Sampled Novel Views

Figure A4: **Continuous rotation examples on SUNRGB-D**. We rotate the camera around the multi-object composites, successfully synthesizing plausible novel-view images across a wide range of camera pose variations.

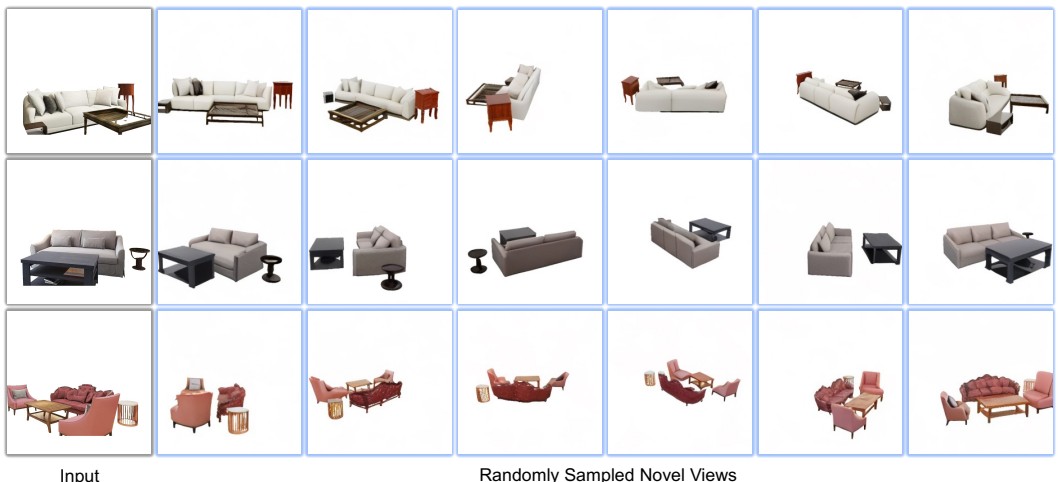

Input                                                    Randomly Sampled Novel Views

Figure A5: **Continuous rotation examples on 3D-FRONT**.

### B.6    MUTUAL OCCLUSION

In multi-object compositions, mutual occlusion between objects is very common. Although we did not specifically design the method to make the model aware of mutual occlusion, the model has learned some understanding of these occlusion relationships. A series of research efforts (Van Hoorick et al., 2023; Ozguroglu et al., 2024; Xu et al., 2024; Zhan et al., 2024; Zhu et al., 2017; Zhan et al.,

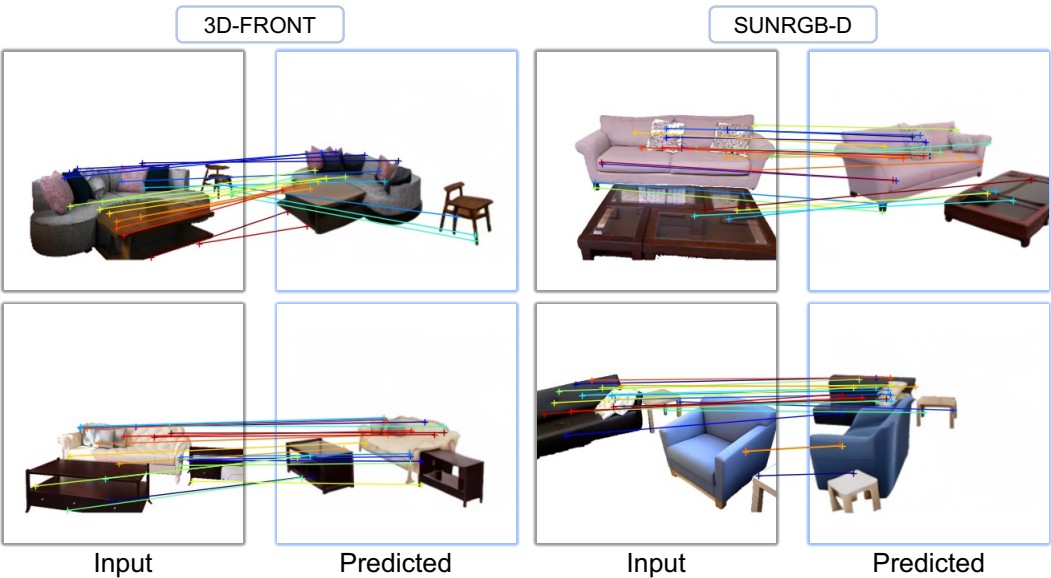

Figure A6: **Visualized cross-view matching results**. Since we do not have ground truth image for 3D-FRONT and SUNRGB-D, we only visualize cross-view matching results using our predicted images. But we can still observe a strong cross-view consistency from the accurate matching results.

Table A3: **Ablation study on various datasets.**

| Dataset | Method | Novel View Synthesis | | | Placement | Cross-view Consistency | |
| --- | --- | --- | --- | --- | --- | --- | --- |
| | | PSNR(↑) | SSIM(↑) | LPIPS(↓) | IoU(↑) | Hit Rate(↑) | Dist(↓) |
| Room-Texture | w/o depth | 9.829 | 0.705 | **0.365** | **25.7** | 5.5 | **75.3** |
| | w/o mask | 9.576 | 0.699 | 0.384 | 24.2 | 2.7 | 92.2 |
| | w/o sch. | 9.173 | 0.689 | 0.392 | 22.4 | 2.3 | 88.6 |
| | Ours | **10.014** | **0.718** | 0.366 | 24.2 | **6.1** | 78.1 |
| Objaverse | w/o depth | 17.457 | 0.835 | 0.178 | 50.5 | 23.0 | 52.6 |
| | w/o mask | 17.176 | 0.834 | 0.187 | 47.3 | 11.1 | 57.1 |
| | w/o sch. | 16.642 | 0.825 | 0.210 | 43.2 | 6.3 | 55.0 |
| | Ours | **17.749** | **0.840** | **0.169** | **51.3** | **50.0** | **47.2** |

2020) specifically focus on addressing mutual occlusion relationships by predicting the amodal masks or synthesizing amodal appearance, but these models typically do not consider scenarios involving camera view change. Moreover, there may not be a well-established metric to measure how well the model understands mutual occlusion from novel viewpoints. We provide a simple experiment and discussion in this section to illustrate model's comprehension of mutual occlusion.

First, in the context of novel view synthesis, the comprehension of occlusion relationships can be divided into two parts. The first is the ability to synthesize parts that were occluded in the input view. The second is the ability to synthesize new occlusion relationships under the novel view. We show several examples of synthesizing occluded parts and synthesizing new occlusions in Fig. A7. We believe this capability is learned in a data-driven way since the multi-object composites are physically plausible regarding these occlusion relationships.

Secondly, we now propose a new metric to evaluate the capability of understanding mutual occlusion under this setting. We first use visible mask and amodal mask in the input-view image to determine how heavily an object is occluded:

1. If an object's visible mask is exactly its full mask, there exists no occlusion.
2. If an object's visible mask is more than 70% of its full mask, the object is occluded.
3. If an object's visible mask is less than 70% of its full mask, the object is heavily occluded.

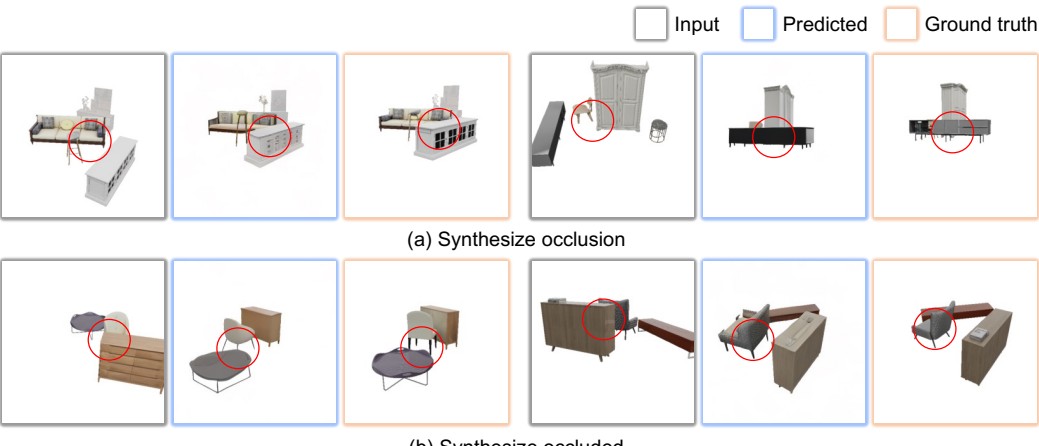

Figure A7: **Occlusion Synthesis Capability**. Our proposed method can synthesize new occlusion relationship under novel views as shown in the highlighted area of sofa or cabinet in (a). Our method can also hallucinate occluded parts as shown in the highlighted area of chairs in (b).

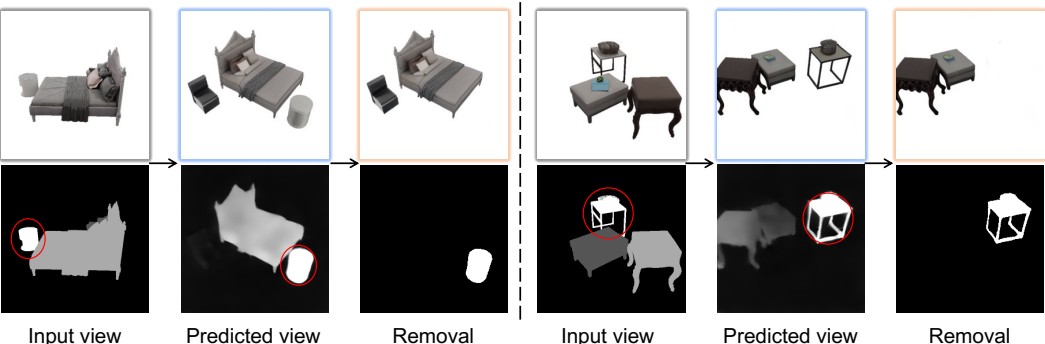

Figure A8: **Object Removal Example**. We can remove an object under novel views by setting a threshold to the predicted mask image and delete corresponding pixels.

Afterward, we segment the predicted view image with ground truth per-object visible mask. We calculate the specific region's PSNR, SSIM, and LPIPS metrics as shown in Tab. A4. It can reflect how well our model and baseline models are at synthesizing novel view plausible images that are originally occluded under the input view. There are 10903 fully visible objects, 6058 occluded objects, and 2215 heavily occluded objects. This experiment is conducted on our own C3DF.

Table A4: **Evaluation on objects with varying extents of occlusion.**

| Method | Visible | | | Occluded | | | Heavily Occluded | | |
|---|---|---|---|---|---|---|---|---|---|
| | PSNR(↑) | SSIM(↑) | LPIPS(↓) | PSNR(↑) | SSIM(↑) | LPIPS(↓) | PSNR(↑) | SSIM(↑) | LPIPS(↓) |
| Ours | **11.45** | **0.56** | **0.13** | **11.33** | **0.55** | **0.14** | **10.57** | **0.55** | **0.14** |
| Zero-1-to-3 | 9.46 | 0.54 | 0.16 | 9.33 | 0.52 | 0.17 | 9.00 | 0.53 | 0.16 |
| Zero-1-to-3[†] | 9.68 | 0.55 | 0.14 | 9.54 | 0.52 | 0.15 | 9.26 | 0.53 | 0.15 |

## C FAILURE CASES AND LIMITATIONS

**Failure Cases**   We showcase two failure cases in Fig. A10. We can observe that delicate structure and texture like colorful pillows on the sofa or slim legs of chairs are hard for our model to learn. Though object placement is approximately accurate, more fine-grained consistency is not quite ideal

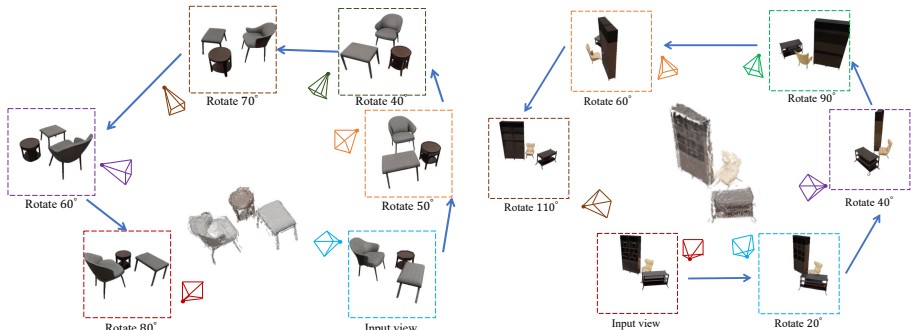

Figure A9: **Reconstruction results using DUSt3R.** We rotate our camera around the multi-object composite and use the predicted images along with the input-view image for reconstruction.

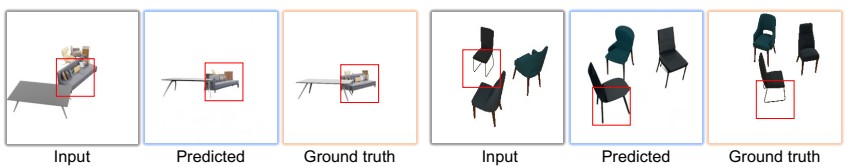

Figure A10: **Failure Cases**. It is hard for our model to learn extremely fine-grained consistency on objects with delicate structure and texture.

in these cases. We believe training with a higher resolution and incorporating epipolar constraints will mitigate this problem in the future.

**Limitations**    We identify two limitations of our work. Firstly, though we achieve stronger cross-view consistency with the input view image, our model does not guarantee the multi-view consistency between our synthesized images. It is plausible to synthesize any results in areas with ambiguity, leading to potential multi-view inconsistency in our model. We believe incorporating multi-view awareness techniques Shi et al. (2023b); Wang & Shi (2023); Shi et al. (2023a); Kong et al. (2024); Liu et al. (2023d); Yang et al. (2024a) can mitigate this problem. Secondly, we do not model background texture in our framework due to difficulty of realistically mimicking real-world background texture, making it less convenient to directly apply our method to in-the-wild images. We believe training on more realistic data with background in the future can make our model more convenient to use.

## D  POTENTIAL NEGATIVE IMPACT

The use of diffusion models to generate compositional assets can raise ethical concerns, especially if used to create realistic yet fake environments. This could be exploited for misinformation or deceptive purposes, potentially leading to trust issues and societal harm. Additionally, hallucinations from diffusion generation models can produce misleading or false information within generated images. This is particularly concerning in applications where accuracy and fidelity to the real world are critical.

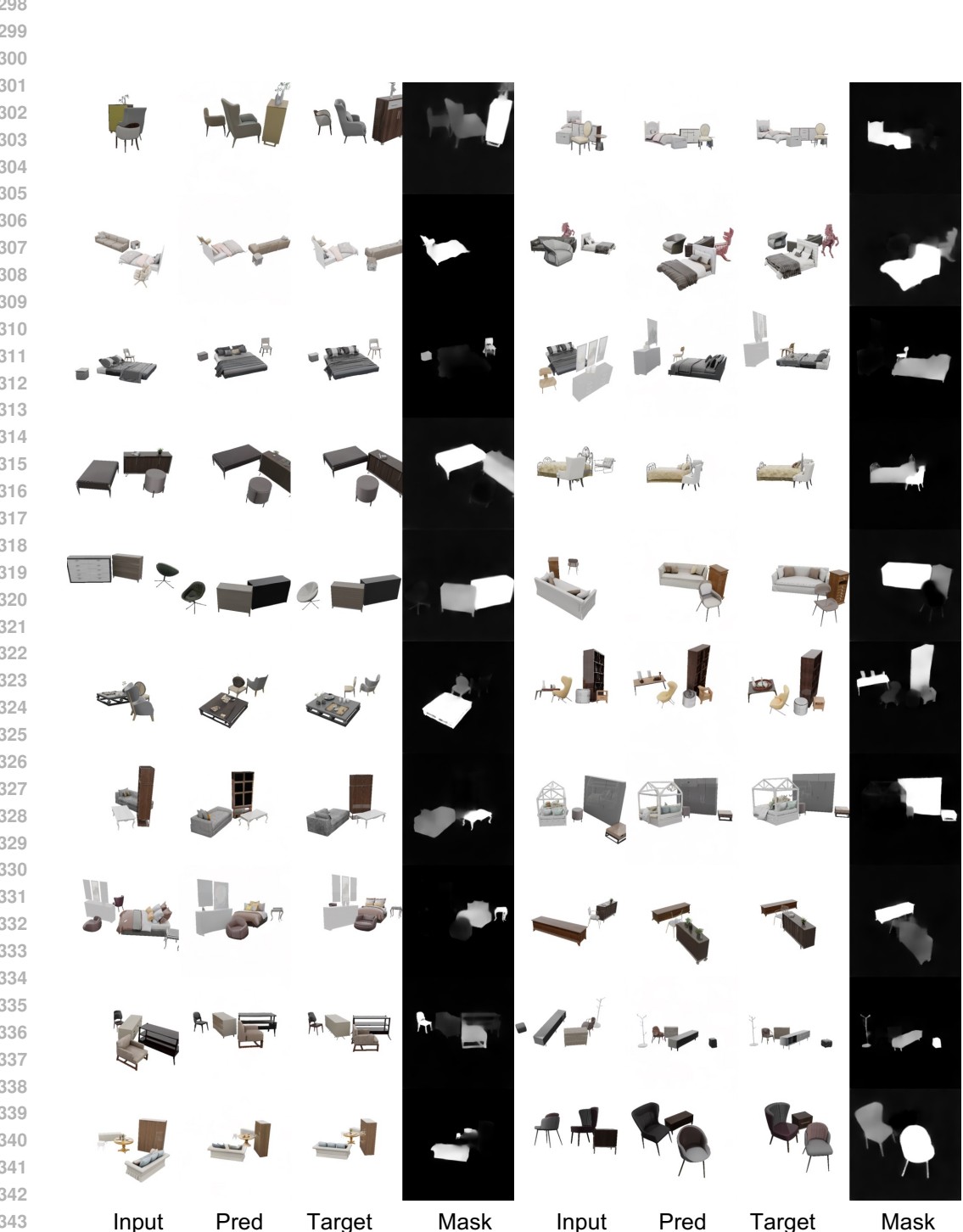

Input    Pred    Target    Mask    Input    Pred    Target    Mask

Figure A11: **More visualized results on C3DFS dataset.**

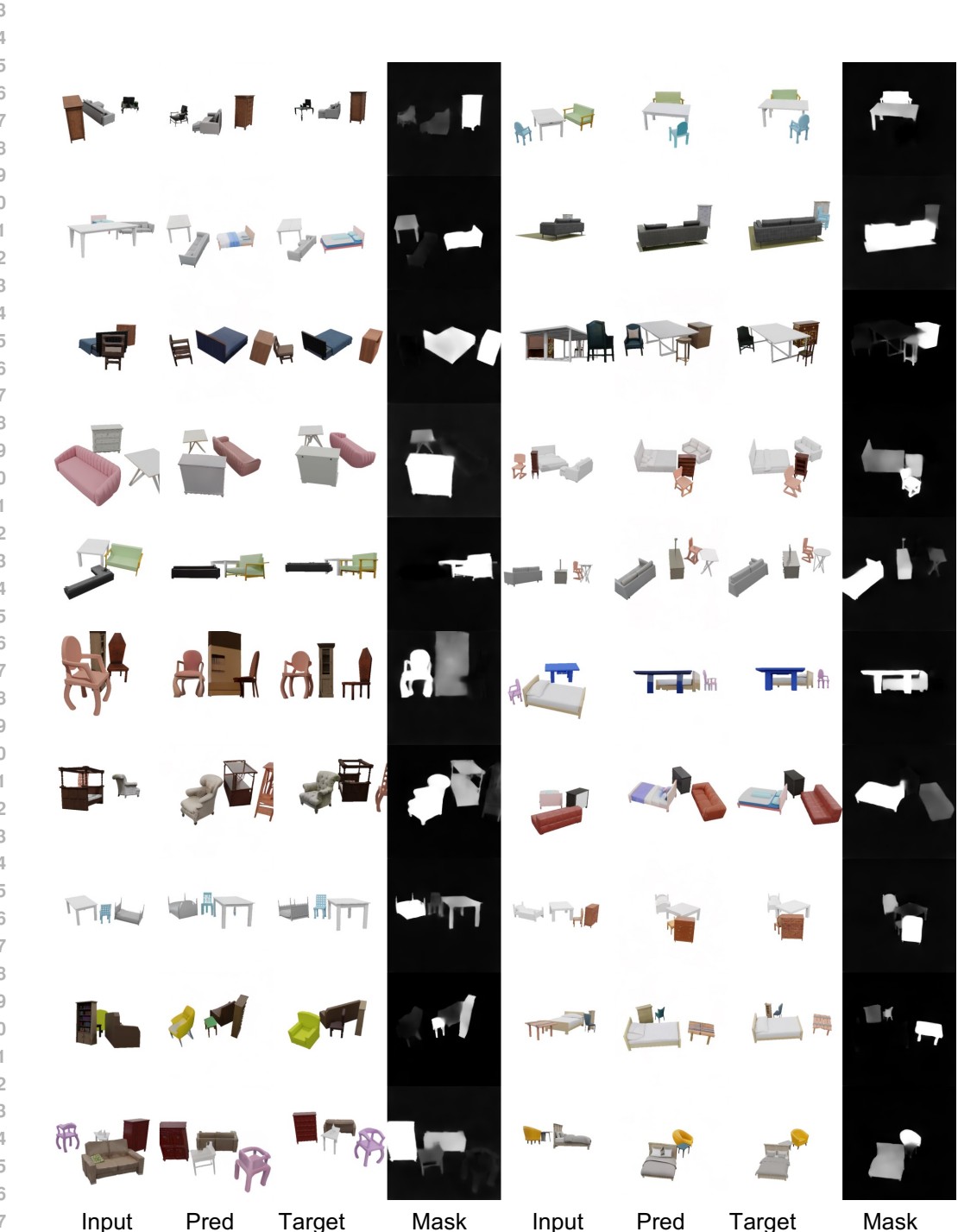

Input    Pred    Target    Mask    Input    Pred    Target    Mask

Figure A12: **More visualized results on Objaverse dataset.**

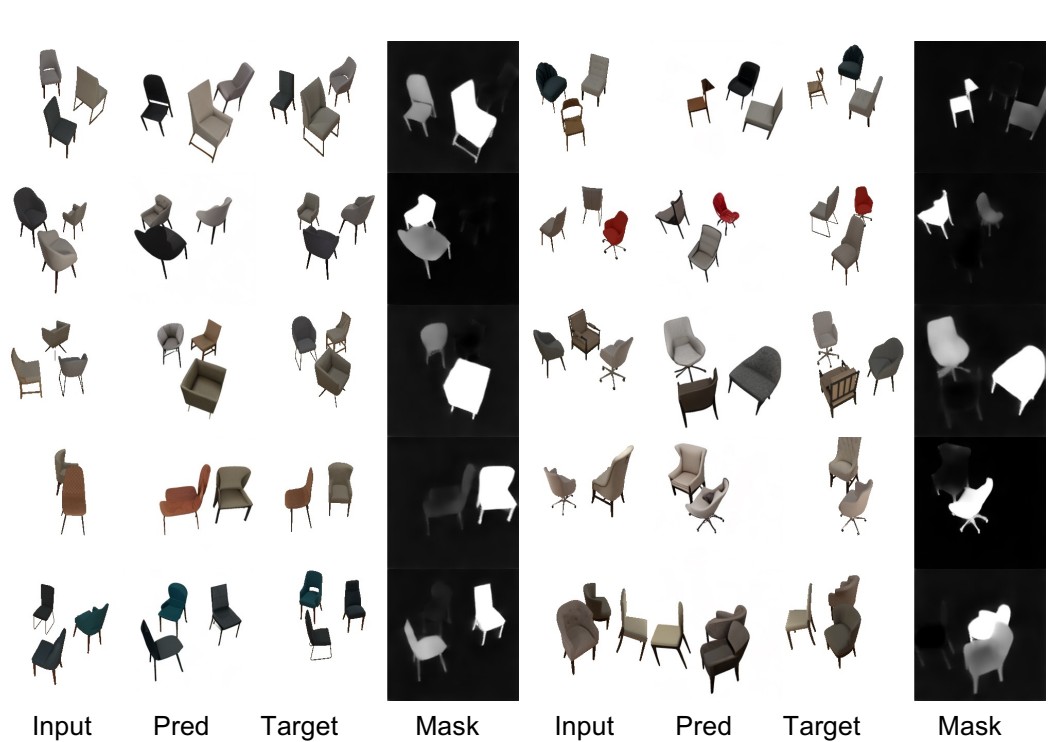

Input    Pred    Target    Mask    Input    Pred    Target    Mask

Figure A13: **More visualized results on Room-Texture dataset.**

