# OpenReview forum: "MOVIS: Enhancing Multi-Object Novel View Synthesis for Indoor Scenes"
_ICLR.cc/2025/Conference — ICLR 2025 Conference Withdrawn Submission_

### Official Review · Reviewer_PBVG · 2024-11-02

**Soundness:** 2
**Presentation:** 2
**Contribution:** 2
**Rating:** 5
**Confidence:** 5

**Summary:**

Major points of the paper:

Structure-Aware Features in NVS Models: MOVIS enhances the model's understanding of spatial structure by integrating depth and object mask features, aiding in object recognition and spatial arrangement. Ablation studies on the C3DF dataset show that removing depth or mask features reduces placement accuracy, as demonstrated by lower Intersection over Union (IoU) and Hit Rate​

Auxiliary Task for Mask Prediction: MOVIS introduces an auxiliary task requiring the model to predict novel view object masks, enhancing object differentiation and placement in multi-object settings. Results indicate that including this auxiliary task significantly improves object placement and cross-view consistency, particularly reflected in higher IoU and Hit Rate on the C3DF test​.

Structure-Guided Timestep Sampling Scheduler: A novel scheduler optimizes global object placement early in training and focuses on local details later.

**Strengths:**

From the paper, it includes following pros.
- The method provide Enhanced Spatial Understanding: Incorporating depth and object mask data enhances spatial awareness, critical in multi-object scenes.
- It Improves Cross-View Consistency: The model demonstrates strong consistency across views, validated by high Hit Rate and lower matching distance on datasets like Objaverse.
- It is Generalized to Unseen Data: MOVIS shows strong adaptability across datasets, including synthetic and real-world scenes​.

**Weaknesses:**

However, there are few problems, first it incorporating additional structure-aware inputs could require independent models for different modalities, including a depth estimation network or a segmentation network. In the paper, it seems the ground truth has been

Also, the experimenting datasets are mostly synthetic data, while it might also worth to check the performance on real object dataset or random generated images, like MVImagenet etc.

For the Scheduler Strategy, it seems the strategy is specifically tuned with respect to the multi-object synthetic dataset, might require dataset-specific calibration, where the generalization of such strategy is limited.  In general, the learning process is automatically figured during the training where simple structures are learned first and then details.  A manually tuned \mu(s) induce extra hyperparameters which might not be generalizable. Should Eqn.(7) indicate higher timestep at begining while lower at the later stage, while it seems current written format is reversed.

**Questions:**

Depth and Mask Input Dependency: Given the reliance on structure-aware features (depth and object masks), how does MOVIS handle scenes where depth or object mask predictions are imperfect? Is there a degradation in performance, and if so, are there any robustness tests to quantify this?   The paper does not appear to include experiments where depth or mask estimation networks replace the input in the MOVIS method. Testing with depth or mask estimations from a network could help assess the robustness of MOVIS under real-world conditions where perfect structural data may not be available.

Timestep Scheduler Fine-Tuning: does it also work with other dataset, or other methods such as zero123 etc ? Does the scheduler only work for multi-object indoor scene senario ?

---

### Official Review · Reviewer_WrMB · 2024-11-02

**Soundness:** 3
**Presentation:** 3
**Contribution:** 2
**Rating:** 5
**Confidence:** 4

**Summary:**

The paper proposes a novel approach called MOVIS to enhance structural awareness in view-conditioned diffusion models for multi-object novel view synthesis (NVS).

MOVIS aims to address the limitations of existing single-object NVS methods when applied to multi-object scenarios, which often result in incorrect object placement and inconsistent shape and appearance under novel views. The method integrates structure-aware features such as depth and object masks into the denoising U-Net to improve object comprehension and spatial relationship awareness. Additionally, an auxiliary task is introduced where the model predicts novel view object masks to further enhance object differentiation and placement. A structure-guided timestep sampling scheduler is also proposed to balance learning between global object placement and fine-grained detail recovery during training.

Extensive experiments on synthetic and real-world datasets demonstrate MOVIS’s superior performance in consistent object placement, shape, and appearance, as well as its strong generalization to unseen datasets.

**Strengths:**

- MOVIS integrates structure-aware features and an auxiliary task for novel view mask prediction, improving the model’s comprehension of spatial relationships and accurate object placement in multi-object scenarios.

- MOVIS demonstrates generalization across unseen datasets and employs a structure-guided timestep sampling scheduler that balances global object placement with fine-grained detail recovery, enhancing overall synthesis quality.

**Weaknesses:**

- This paper has significant issues in presenting results. Although the paper claims to achieve consistent multi-view synthesis for multiple objects, the results do not support this claim.

- For example, in Figure 4, the third input (Refer to row 3 column 1 / and row 3 column 3) shows an orange and a yellow pillow, which are distinctly different in color. However, the output shows two yellow pillows being generated. Additionally, the sofa in the generated result includes unfounded noise (on the right side of the sofa), and the thickness of the desk legs has also changed. Furthermore, the consistency of the results presented in this paper is not well maintained, as can be observed in various details, such as the color of the chairs.

- While it is commendable that the paper attempts to address issues that previous models have not solved, these problems contradict the statement in the Abstract mentioning "especially incorrect object placement and inconsistent shape and appearance under novel views."

**Questions:**

Mentioned in the weaknesses section. It would be helpful if the authors could clearly address these mentioned points during the rebuttal period.

---

### Official Review · Reviewer_yEqj · 2024-11-03

**Soundness:** 3
**Presentation:** 3
**Contribution:** 2
**Rating:** 5
**Confidence:** 4

**Summary:**

This paper introduces a method to improve cross-view consistency in multi-object NVS using pre-trained diffusion models. MOVIS enhances model inputs with depth and object mask information, adds an auxiliary task for predicting object masks in new views, and uses a structure-guided timestep sampling scheduler to improve object placement and detail recovery.

**Strengths:**

1. Targeting multi-object NVS is interesting, as most current NVS methods are validated on single-object scenarios. By addressing challenges like correct object placement, shape, and appearance across views, this direction opens up new possibilities for compositional scene generation.

2. Using corresponding points as a validation metric is a valuable addition, as it provides a more robust measure of cross-view consistency. Unlike image-level metrics alone, corresponding points offer a way to directly assess if objects maintain accurate positions, shapes, and appearances across different viewpoints. This metric helps validate the model’s effectiveness in capturing structural relationships and ensures synthesized images maintain coherent spatial relationships across views.

**Weaknesses:**

1. The paper lacks commonly adopted NVS evaluation metrics that directly assess multi-view consistency, such as running 3D reconstruction (like nerf) to compute mesh differences (e.g., Chamfer distance) or re-rendering metrics like SSIM or LPIPS. These approaches provide a more direct evaluation of the generated multi-view consistency by quantifying structural and perceptual alignment across views. Without these metrics, it’s harder to objectively compare the quality of synthesized outputs.

2. While the paper references several recent improvements on Zero123, including methods like SyncDreamer and Consistent123, it doesn’t experimentally compare MOVIS with these state-of-the-art techniques. Given that these methods also tackle cross-view consistency NVS challenges, a direct comparison would provide valuable insights into how MOVIS performs relative to these advances, strengthening the claims about its effectiveness.

3. The paper could improve fairness in comparisons by controlling variables like training data and strategy, which currently differ from the baseline methods. Using consistent training data and experimental protocols would ensure a fair comparison. A suggested approach includes: 1) selecting a subset evaluation dataset from the Objaverse validation set with multiple objects; 2) training MOVIS on this dataset using its proposed strategy; and 3) evaluating MOVIS and baseline methods on this subset. This setup would yield more robust and convincing results by isolating the effects of MOVIS’s architectural and training improvements.

**Questions:**

1. In Table 1, finetuning Zero123 on the 3D-FUTURE dataset seems to unexpectedly worsen its performance, which is unusual as fine-tuning typically enhances or maintains model quality. Could the authors elaborate more on their fine-tuning process on the new dataset? This result may indicate potential overfitting, which is often avoidable with techniques like regularization, LoRA, data augmentation, or early stopping. For a fairer comparison, consider integrating 3D-FUTURE into the original training data (Objaverse-10M and Objaverse-800K) and training Zero123 jointly. This approach would help assess the true benefit of adding 3D-FUTURE and prevent overfitting by diversifying training instances across datasets.

2. For the Zero123 comparisons, is the Zero123-XL version being used? This clarification is important, as the standard and XL versions have different capabilities and model sizes, which could influence performance. Comparing MOVIS to Zero123-XL, if it’s not already the case, would provide a stronger benchmark and ensure that results accurately reflect improvements attributable to MOVIS rather than differences in model scale.

---

### Official Review · Reviewer_knKm · 2024-11-07

**Soundness:** 2
**Presentation:** 3
**Contribution:** 2
**Rating:** 3
**Confidence:** 5

**Summary:**

This paper presents MOVIS, an approach designed to enhance multi-object novel view synthesis (NVS) from single images by addressing the limitations of existing methods that primarily focus on single-object scenarios. MOVIS incorporates structure-aware features, such as depth and object masks, into the model's inputs to improve the understanding of object instances and their spatial relationships. Additionally, it introduces an auxiliary task for predicting novel view object masks, which aids in accurate object differentiation and placement. A structure-guided timestep sampling scheduler is also proposed to balance the learning of global object placement and fine-grained detail recovery during training. Extensive experiments demonstrate that MOVIS achieves good generalization capabilities and produces novel view syntheses, highlighting its potential to advance 3D-aware multi-object NVS tasks in complex environments.

**Strengths:**

- The incorporation of structure-aware features, such as depth and object masks, enhances the model's ability to understand complex spatial relationships in multi-object scenarios.

- The introduction of an auxiliary task for predicting novel view object masks demonstrates a thoughtful approach to improving model performance.

**Weaknesses:**

- The paper propose a new scheduler, the structure-guided timestep sampling scheduler. This new scheduler is motivated by [1] and further improve it by, not using a fixed variance when sampling the timestep value from a gaussian distribution, but use a linear decay from variance = 1000 to variance = 500. I have two concerns with this new strategy:
  1) While this idea sounds more reasonable than [1], I did not find an ablation study to show it better than [1]. Table 2 seems to have some numbers reported, but it shows the comparison between w/ and w/o this scheduler, rather than the comparison between the proposed scheduler and the scheduler in [1].
  2) The idea of [1], using a gaussian sampling strategy to replace the original uniform sampling strategy, is motivated by the fact that the pretrained stable diffusion already learns the detail very well. Therefore the smaller timestep value is less likely to be sampled. If the proposed new scheduler is designed to sample more from larger timestep value in the initial training stage, then followed by the smaller timestep value in the late training stage, isn't that the same as the original uniform sampling strategy? The larger and smaller timestep value get equal chance to be sampled regarding the entire training process, which is essentially the "uniform".


- While the multi-view consistency issue is becoming the key of evaluating the performance of single-image novel-view synthesizer, state-of-the-art methods [2,3,4] has started to use mutliview-attention and gains notable improvements. The proposed method is not compared with those approaches and it is not convincing to conclude that the proposed module can performs better than these state-of-the-art method.

Reference:

[1] Efficient-3DiM: Learning a Generalizable Single-image Novel-view Synthesizer in One Day.

[2] CAT3D: Create Anything in 3D with Multi-View Diffusion Models.

[3] Zero123++: a Single Image to Consistent Multi-view Diffusion Base Model.

[4] MVDream: Multi-view Diffusion for 3D Generation

**Questions:**

- In Table 1, why the PSNR value in Room-Texture dataset is very low? I assume PSNR lower than 10 shows the produced images contains entire noise with zero meaningful items.

---

### Note · Authors · 2024-11-15

I have read and agree with the venue's withdrawal policy on behalf of myself and my co-authors.